# Mitochondrial DNA Copy Numbers and Lung Cancer: A Systematic Review and Meta-Analysis

**DOI:** 10.3390/ijms26146610

**Published:** 2025-07-10

**Authors:** Manuela Chiavarini, Jacopo Dolcini, Giorgio Firmani, Kasey J. M. Brennan, Andrès Cardenas, Andrea A. Baccarelli, Pamela Barbadoro

**Affiliations:** 1Department of Health Sciences, University of Florence, Viale GB Morgagni 48, 50134 Florence, Italy; 2Department of Biomedical Sciences and Public Health, Section of Hygiene, Preventive Medicine and Public Health, Polytechnic University of the Marche Region, 60126 Ancona, Italy; 3Department of Environmental Health, Harvard T.H. Chan School of Public Health, Boston, MA 02115, USA; 4Department of Epidemiology and Population Health, Stanford University, Palo Alto, CA 94305, USA

**Keywords:** lung cancer, mitochondrial DNA, blood, sputum, meta-analysis, systematic review, public health

## Abstract

Lung cancer (LC) remains the leading cause of cancer-related deaths worldwide, representing a major public health challenge. Mitochondria, due to their essential role in cell survival and death, have become the object of increasing research attention, and mitochondrial DNA copy number (mtDNAcn) alterations have been investigated as potential biomarkers of LC risk. This systematic review and meta-analysis evaluated studies exploring the relationship between the mtDNAcn in biological samples (e.g., blood, sputum) and LC risk. A systematic search on PubMed, Web of Science, and Scopus identified five eligible studies. Using a random-effects model, the pooled analysis did not detect a significant association between the mtDNAcn and LC risk, independently of the study design. Furthermore, no significant publication bias was detected. These results highlight the need for further studies to better understand the role of the mtDNAcn in the development of LC and to explore potential confounders influencing this relationship.

## 1. Introduction

LC is the leading diagnosed cancer in men and the second most common in women. In 2023, it was been estimated that 1.96 million people will be diagnosed with LC, and 609,820 people will die from the disease [1]. LC has two primary pathological classifications: small-cell lung cancer (SCLC) and non-small cell lung cancer (NSCLC). Notably, NSCLC represents about 85% of all newly diagnosed LC cases, making it the predominant pathological subtype [2,3,4]. Cigarette smoking is the leading risk factor for LC, responsible for 80% of cases and deaths associated with the disease [5]. Effective tobacco control policies and regulations have prevented the disease by lowering incidence and mortality rates [6,7]. Currently, the primary treatment approaches for LC include surgery, radiation therapy, chemotherapy, innovative molecular targeted therapies, and immunotherapy. However, these approaches often fail to achieve the desired therapeutic results in LC patients [8,9]. Moreover, the socio-economic burden of cancer is significant and includes not only treatment costs but also expenses related to lost productivity, disability, and premature death [6]. Environmental and lifestyle factors, together with genetic factors, contribute to the onset and progression of cancer, and there is an urgent need for prevention tools and strategies to better assess and stratify the possible risk of exposed subjects [10,11]. New minimally invasive biomarkers involving blood or sputum sampling have shown promising results in terms of prevention, diagnosis, and follow-up [12,13,14]. Among possible interesting candidates in terms of the risk evaluation of LC, the mitochondrial DNA (mtDNA) and specifically the mtDNAcn (mtDNAcn) has been recently investigated [15,16].

It is well known that mitochondria play a central role in cellular energy production, the control of metabolic stress, and the maintenance of cellular homeostasis, while also inducing reactive oxygen species (ROS) involved in multiple diseases [17,18]. The number of mitochondria and mitochondrial DNA (mtDNA) cells may differ based on their specific functions [19]. Recent studies have explored the role of genetic and epigenetic markers in LC risk, with a focus on the potential of methylation biomarkers as indicators of disease susceptibility and early diagnosis [20,21]. Furthermore, a meta-analysis has highlighted the influence of the leukocyte telomere length on LC susceptibility [22]. Regarding cancer, an increasing number of studies that measured mtDNAcn from blood between cancer patients and healthy controls have identified associations between mtDNAcn levels and increased risks of cancer, finding an association with renal cell carcinoma, pancreatic cancer, and colon–rectal cancer [23,24,25]. Concerning cancer’s beginning and development, several molecular pathways have been shown to have a possible association between mtDNAcn alterations and cancer diseases: Alterations in mtDNAcn may reflect oxidative damage, impaired mitophagy, or the dysregulation of mitochondrial–nuclear cross-talk, processes known to influence tumorigenesis. Reactive oxygen species (ROS) generated during mitochondrial respiration can cause mutations in both nuclear and mitochondrial genomes, promoting oncogenic transformations. Furthermore, changes in mtDNAcn may modulate key signaling pathways, including HIF-1α, NF-κB, and p53, thereby influencing cell proliferation, inflammation, and apoptosis. Understanding the molecular implications of altered mtDNAcn in peripheral tissues could provide novel insights into the early mechanisms underpinning cancer development and its potential use in minimally invasive screening, including for LC.

In recent years altered mtDNAcn has emerged as a biomarker of respiratory diseases and it has been found to be linked to several lung diseases such as asthma [26], chronic obstructive pulmonary disease (COPD) [27], and LC [28,29]. In contrast, some studies concluded that mtDNAcn in peripheral blood predicted a poor cancer diagnosis, while better outcomes were obtained from measures coming from tumor tissue [30], and others did not find a consistent association between the prediagnostic mtDNAcn levels and LC risk across several populations [31], showing a lack of general of consensus on this association. To the best of our knowledge, there are no systematic reviews and meta-analyses that investigate the association of mtDNAcn obtained by peripheral sampling methods, such as blood or saliva, and LC risk; therefore, the aim of our systematic review and meta-analysis is to evaluate the association between the mtDNAcn and LC incidence.

## 2. Materials and Methods

The present meta-analysis was conducted in accordance with the MOOSE (Meta-Analysis of Observational Studies) guidelines [32] and the PRISMA statement [33].

This study protocol was registered in the International Prospective Register of Systematic Reviews (www.crd.york.ac.uk/PROSPERO/, registration No: CRD42024516565) (accessed on 22 February 2024).

### 2.1. Search Strategy and Data Source

We conducted a systematic literature search up to 3 June 2025 through PubMed http://www.ncbi.nlm.nih.gov/pubmed/ (accessed on 4 June 2025), Web of Science http://wokinfo.com/ (accessed on 4 June 2025), and Scopus https://www.scopus.com/ (accessed on 4 June 2025) databases to find original articles discussing the association between changes in mtDNAcn and LC.

The following keywords were used: (mitochondrial DNA copy number) AND (lung cancer risk).

The different associations of keywords combined with Boolean operators used for each database are shown in Appendix A.

We did not apply any date limitation but, due to translation restrictions, we opted to include only English-language studies.

Moreover, to include possible relevant publications, we manually examined the reference lists of the included articles and possible new relevant reviews.

### 2.2. Eligibility Criteria

Among the selected studies, only those that satisfied the following conditions were considered eligible: (i) the investigation focused on the association between mtDNAcn alterations and lung cancer (LC); (ii) the study design was observational, specifically case–control, cohort, or cross-sectional; and (iii) the effect size was expressed through odds ratios (ORs), relative risks (RRs), or hazard ratios (HRs), all accompanied by 95% confidence intervals (CIs).

Studies combining LC with other cancer types were excluded from the analysis. Each potentially eligible article underwent independent screening and quality evaluation by two reviewers. Any discrepancies were discussed and resolved either through consensus or by consulting a third reviewer. Narrative reviews and meta-analyses were excluded, despite their utility for contextual understanding. Importantly, no study was excluded solely due to methodological limitations or data quality concerns.

### 2.3. Data Extraction and Quality Assessment

Two reviewers independently performed data extraction and quality assessment. Disagreements were resolved by consensus; a senior researcher was consulted when necessary to reach a final decision. For each selected study, we extracted the following information: first author’s last name, year of publication, country, study design, sample size, population characteristics (sex, age, race, BMI, smoking status), duration of follow-up for cohort studies, risk estimates with 95% CIs, type of LC, tissue type, and confounding factors adjustment. When multiple estimates were reported in the article, we extracted those adjusted for the most confounding factors. The Newcastle–Ottawa Scale (NOS) was used to assess the quality of the literature for cohort and case–control studies using a 9-star system, as shown in Appendix A. The full score was 9 and a total score ≥ 7 was used to indicate a high-quality study [33].

### 2.4. Statistical Analysis

The overall effect size statistic was estimated considering the risk of LC associated with the highest versus the lowest level of changes in mtDNAcn. In case of multivariable models, we selected the risk values coming from those that included the greatest number of potential confounding variables. To account for the high heterogeneity, a random-effects model and inverse variance weighting methods were used to calculate the sum of OR and the 95% confidence intervals.

To consider an effect statistically significant, a cut-off of a two-tailed *p* < 0.05 was applied. The chi-square-based Cochran’s Q statistic and the I^2^ statistic were used to evaluate heterogeneity in results across studies [34]. The I^2^ statistic yields results ranging from 0% to 100% (I^2^ = 0–25%, no heterogeneity; I^2^ = 25–50%, moderate heterogeneity; I^2^ = 50–75%, large heterogeneity; and I^2^ = 75–100%, extreme heterogeneity) [35]. The results of the meta-analysis may be biased if the probability of publication is dependent on the study results.

We used the methods of Begg and Mazumdar [36] and Egger et al. [37] to detect publication bias. Both methods tested for funnel plot asymmetry, with the former being based on the rank correlation between the effect estimates and their sampling variances and the latter on a linear regression of a standard normal deviation on its precision.

If a potential bias was detected, we further conducted a sensitivity analysis to assess the strength of combined effect estimates and the possible influence of the bias and to have the bias corrected. We also conducted a sensitivity analysis to investigate the influence of a single study on the overall risk estimate, by omitting one study in each turn. We considered the funnel plot to be asymmetrical if the intercept of Egger’s regression line deviated from zero, with a *p*-value < 0.05.

The analyses were performed using the ProMeta 3 (Prometa S.R.L., Torviscosa, UD, Italy) statistical program, and the calculations on data extracted from the original papers were performed using STATA 15 (Stata Corp., College Station, TX, USA).

## 3. Results

### 3.1. Study Selection

The database search identified 20 records from PubMed, 52 from Web of Science, and 29 from Scopus. After eliminating duplicates (n = 31), a total of 68 unique articles remained and were assessed based on their titles and abstracts. Of these, 52 publications were excluded because they consisted of reviews, meta-analyses, pooled studies, commentaries, or case reports (Figure 1).

A total of 16 studies proceeded to the full-text evaluation. Following this step, 11 additional articles were excluded because they did not align with the predefined inclusion criteria. Specifically, they involved research on genetic polymorphisms, animal models, or Mendelian randomization. Ultimately, five studies met all eligibility requirements and were included in the final systematic review and meta-analysis [31,38,39,40,41].

### 3.2. Study Characteristics and Quality Assessment

The general characteristics of the five studies and relative populations included in the meta-analysis evaluating the association between mtDNAcn and LC risk are shown, respectively, in Table 1.

Studies have been conducted in the United States of America [31,40,41]; Finland [39,40]; and China [38,40]. There were three case–control studies [31,38,39,40,41] and two nested case-control studies [31,40]. Studies have been published from 2009 to 2022, and all of them were carried out in people aged ≥50 years old.

All studies have collected samples from peripheral blood, except one that was collected from sputum [38]. The health outcome investigated was the risk incidence of LC. The study-specific quality scores of selected studies are shown in the last column on the right of Table 1. The quality scores ranged from 6 to 9 (median: 7; mean: 7) in Table 1.

### 3.3. Meta-Analysis mtDNAcn

Concerning the mtDNAcn (Table 2; Figure 2), looking at all observations the overall association between the mtDNAcn and risk of LC was not significant (OR = 0.94; 95% CI: 0.49–1.78); this lack of significance was maintained when excluding Bonner et al. [38] due to the low quality of the study and the different type of samples compared to other articles (OR = 0.81; 95% CI: 0.39–1.67). Stratifying the analyses, LC was not associated with a specific sex or smoking status. Regarding possible heterogeneity, the analysis revealed a significant heterogeneity across the studies. In the combined analysis (ALL), the heterogeneity was high (Q = 132.40, I^2^ = 96.98%, *p* < 0.001), indicating a substantial variability between studies.

### 3.4. Sensitivity Analysis mtDNAcn

Sensitivity analyses investigating the influence of a single study on the LC risk estimates suggested that removing the study by Kennedy et al. [41] resulted in an increment of the LC risk, which was statistically significance (N: 30; OR: 1.16; 95% CI: 1.01, 1.33; *p* = 0.036; I^2^: 41.31, *p* = 0.01; *p* (Egger test): 0.001, *p* (Begg test): 0.002). Excluding Kennedy et al. [41], the publication bias tests became significant because this study specifically examined the difference in risk between quartiles rather than comparing the highest and lowest quartiles, as performed in this systematic review and meta-analysis.

### 3.5. Publication Bias mtDNAcn

No publication bias was detected with Egger’s or Beggs method (Table 2; Figure 3).

### 3.6. Dose–Response

Three articles were included from the dose–response analysis [39,40,41]. The other two studies were excluded because they did not include the division by quartiles [31,38]. The dose–response relationship between the mtDNAcn, expressed as standardized quartiles, and LC risk is presented in Figure 4. On the x-axis the standardized exposure levels corresponding to mtDNAcn quartiles Q1 (reference), Q2, Q3, and Q4 are reported, while on the y-axis the odds ratio (OR) for the LC risk on a linear scale can be found. The red horizontal line at OR = 1 represents the reference category (Q1).

The black solid line shows a slight decrease in ORs across the quartiles, representing a potential decrease in LC risk with an increase in mtDNAcn levels.

The blue dashed lines represent the 95% confidence intervals, which widen at higher quartiles, suggesting an increased uncertainty in the estimates for the upper mtDNAcn quartiles.

The results of the meta-regression analysis showed that the intercept OR was found to be 1.31 (95% CI: 1.02–1.68), indicating that, at the reference level of exposure (standardized exposure = 0, corresponding to the first quartile), the odds of the outcome are approximately 1.31 times higher compared to the baseline odds (OR = 1).

For the standardized exposure, the OR was estimated at 0.91 (95% CI: 0.81–1.03). This suggests that for each one-unit increase in the standardized exposure (i.e., moving from one quartile to the next), there is an approximate 8.9% decrease in the odds of the outcome. Although this trend indicates a reduction in odds with increasing exposure, the 95% confidence interval includes the value of one, suggesting that the association is not statistically significant at the conventional alpha level (*p* = 0.13).

## 4. Discussion

To the best of our knowledge, this is the first systematic review and meta-analysis to summarize the relationship between mtDNAcn and LC. We did not observe statistically significant associations for the overall effect or for the stratifications by the sex and smoking status. However, we consistently observed a trend of an increased risk, with all ORs being greater than one. It is possible that the lack of statistical significance could be due to the limited number of studies and data points, as only five articles were eligible for inclusion in the meta-analysis after our selection process. As previously discussed, there is a growing body of evidence supporting the role of mtDNAcn in the underlying mechanisms of various diseases, including several cancers and aging-related disorders. A common theme in these studies involves anti-inflammatory pathways. The dose–response analysis we performed showed an interesting trend, with decreasing ORs as mtDNAcn levels increased, potentially indicating a role for mtDNAcn in mitigating inflammation and/or the cancer onset. This hypothesis is supported by the fact that the lowest quartile (Q1) of mtDNAcn showed a statistically significant increase in LC risk, but this significance was lost with increasing levels of mtDNAcn, despite the consistently lower OR trend. mtDNAcn is an especially attractive biomarker due to its non-invasive and relatively cost-effective measurement in blood; though its application is limited by several analytical factors affecting accurate and reproducible quantifications [42]. These limitations could have biased the results of our meta-analysis. Moreover, mtDNAcn may be just one factor in a more complex equation, as recent findings suggest that human mtDNA itself can be methylated, influencing the cross-talk between the nucleus and mitochondria and its regulatory control [43].

### Strengths and Limitations

To the best of our knowledge, this is the first study to conduct a systematic review and meta-analysis of mtDNAcn and its association with LC risk, representing a comprehensive, though not exhaustive, approach to better understanding the relationship between this potential biomarker and LC. Moreover, all but one of the included studies were of a high quality, scoring above seven on the Newcastle–Ottawa Scale (NOS), ensuring reliability, even though we acknowledge potential limitations, particularly concerning the inter-rater agreement on the NOS [44,45]. All articles included in this analysis were published within the last decade, reflecting the most up-to-date research in this field. However, several limitations should be noted. First, only five studies were eligible for this systematic review and meta-analysis. This small number of studies may represent limited data, potentially affecting the statistical significance of our results. There was also a lack of standardization across studies regarding the quantification of mtDNAcn levels, as some studies focused on quartiles while others used a continuous classification with absolute numbers. Moreover, the potential influence of the geographic or ethnic variability in the baseline mtDNA copy number cannot be excluded. Although current evidence does not support consistent differences across populations, further studies are needed to clarify whether contextual or population-specific factors may contribute to the observed heterogeneity. Furthermore, considerable heterogeneity was observed across studies, which can be attributed to differences in various analytical factors that affect the accurate and reproducible quantification of mtDNAcn. Additionally, due to limited data availability, the further stratification by the BMI, age, socio-economic status, or other demographic characteristics was not possible.

## 5. Conclusions

In this systematic review and meta-analysis of five studies, mtDNAcn did not show a statistically significant association with LC risk. Despite this result, some aspects, such as the dose–response analysis, revealed interesting findings that could require further investigation. More studies are needed to better standardize the detection of mtDNAcn in both peripheral blood and saliva samples, to allow different studies to be comparable, and especially to identify potential cut-off values for mtDNAcn levels that could help identify physiopathological mechanisms in other areas of the body, such as LC. The scientific community, public health decision-makers, and policymakers need to increase cooperation efforts to improve and enhance the prevention policies, diagnosis, and follow-up of diseases like LC, which remains one of the leading causes of death. The study of mtDNAcn could represent a new and valuable tool, but further knowledge is necessary.

## Figures and Tables

**Figure 1 ijms-26-06610-f001:**
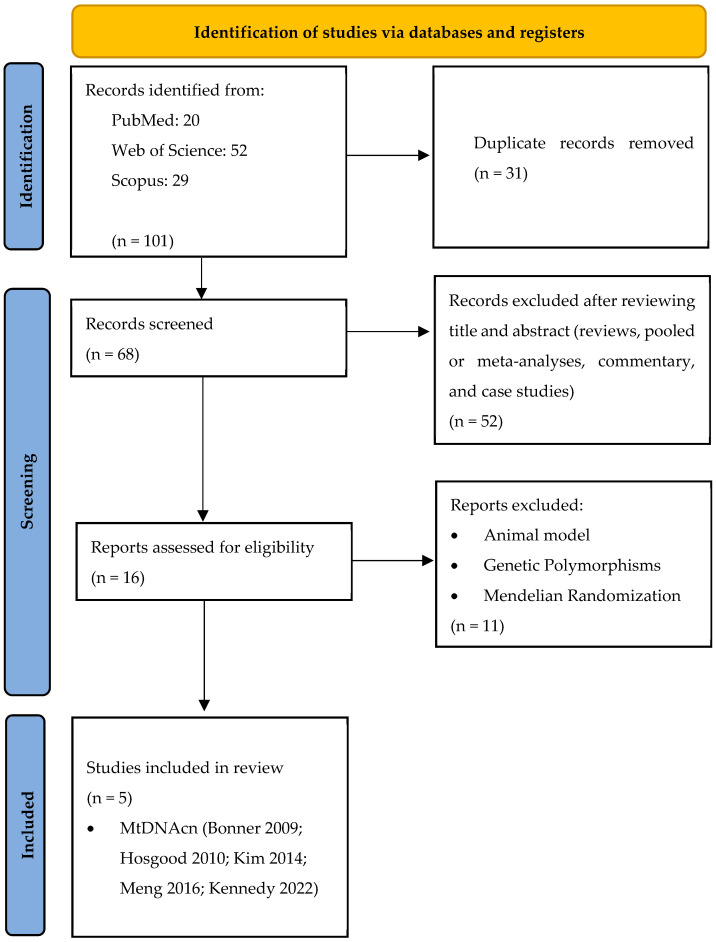
A flow diagram of the systematic literature search on mtDNAcn and LC risk [31,38,39,40,41].

**Figure 2 ijms-26-06610-f002:**
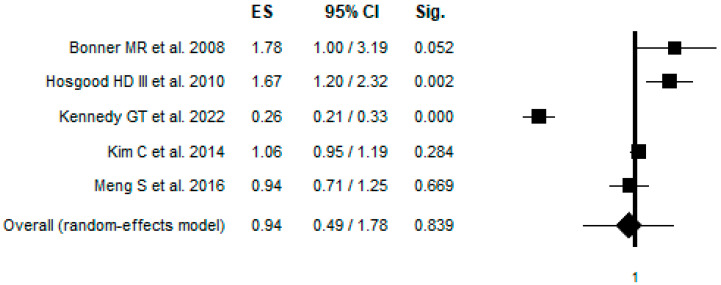
Forest plot of mtDNAcn and risk of LC [31,38,39,40,41].

**Figure 3 ijms-26-06610-f003:**
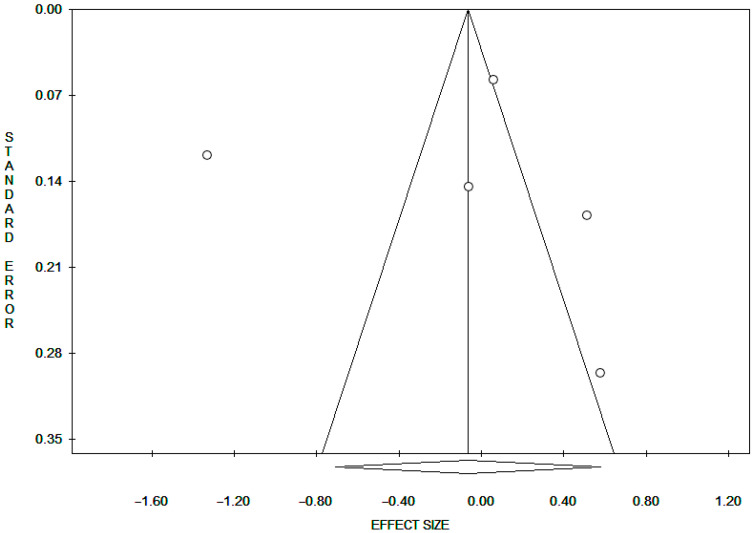
Funnel plot of publication bias of mtDNAcn and risk of LC.

**Figure 4 ijms-26-06610-f004:**
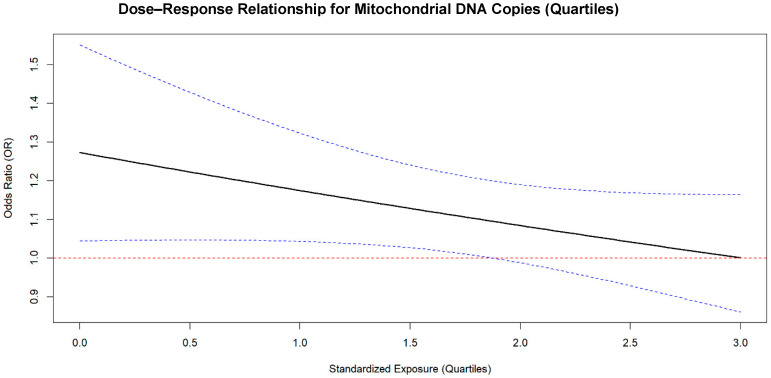
The dose-dependent plot representing the relationship between the mitochondrial DNA copy number (mtDNAcn) and lung cancer (LC) risk. On the x-axis, the standardized exposure levels as mtDNAcn quartiles Q1 (reference), Q2, Q3, and Q4 are reported, while the y-axis represents the odds ratio (OR) for LC risk on a linear scale. The red dashed horizontal line at OR = 1 marks the reference category (Q1). The trend obtained from the meta-regression is shown by the black solid line, showing a mild, non-significant decrease in LC risk with the increase in mtDNAcn. The blue dashed lines are the limits of the 95% confidence intervals, which widen at higher quartiles.

**Table 1 ijms-26-06610-t001:** Characteristics of the studies included in the systematic review and meta-analysis of the association between mtDNAcn and LC risk.

Author, Year, Reference	Cohort ^1^, Location	Study Design	N	Age (M: Mean, Mdn: Median)	Sex (% Male)	Race (% White)	BMI (% or M: Mean, [kg/m^2^] <25)	Type of LC ^2^	Tissue Type	Smoking Status (% Never)	Matched or Adjusted Variables	NOS ^3^
Kennedy GT, 2022 [41]	UPHS, USA	Case–control	Cases: 465 Controls: 378	Cases: 66 (Mdn) Controls: 58 (Mdn)	Cases: 55.7 Controls: 51.3	Cases: 81.3 Controls: 63.8		NSCLC and SCLC	Peripheral blood	Cases: 9.7 Controls: 24.3	Age, race, gender, tobacco use, and tobacco pack-years	8
Meng S, 2016 [31]	NHS, USA	Nested case–control	Cases: 285 Controls: 285	Cases: 59.8 (M) Controls: 59.5 (M)			Cases: 60.9 (%) Controls: 57.3 (%)	LC	Peripheral blood		BMI, physical activity, alcohol consumption. and health eating index score	9
	HPFS, USA	Nested case–control	Cases: 178 Controls: 178	Cases: 66.0 (M) Controls: 65.7 (M)			Cases: 42.9 (%) Controls: 31.8 (%)	LC	Peripheral blood		BMI, physical activity, alcohol consumption, and health eating index score	9
Kim C, 2014 [40]	PLCO, USA	Nested case–control	Cases: 426 Controls: 436	Cases: 64.1 (M) Controls: 63.7 (M)	Cases: 60.8 Controls: 61.2	Cases: 93.7 Controls: 92.7	Cases: 26.8 (M) Controls: 27.4 (M)	LC	Peripheral blood	Cases: 10.8 Controls: 45.4	Age, BMI, pack-years, race, sex, date of enrollment, center (if applicable), and study	7
	SWHS, China	Nested case–control	Cases: 221 Controls: 222	Cases: 59.2 (M) Controls: 59.2 (M)			Cases: 24.6 (M) Controls: 25.0 (M)	LC	Peripheral blood	Cases: 92.3 Controls: 95.0	Age, BMI, pack-years, race, sex, date of enrollment, center (if applicable), and study	7
	ATBC, Finland	Nested case–control	Cases: 227 Controls: 227	Cases: 56.7 (M) Controls: 58.4 (M)			Cases: 25.6 (M) Controls: 26.3 (M)	LC	Peripheral blood	Cases: 0.0 Controls: 0.0	Age, BMI, pack-years, race, sex, date of enrollment, center (if applicable), and study	7
Hosgood HD III, 2010 [39]	ATBC, Finland	Case–control	Cases: 227 Controls: 227	Cases: 58.7 (M) Controls: 58.4 (M)	Cases: 100 Controls: 100		Cases: 25.6 (M) Controls: 26.3 (M)	LAUD and LUSC	Peripheral blood	Cases: 0.0 Controls: 0.0	Age, number of years smoking, and number of cigarettes per day	7
Bonner MR, 2009 [38]	China (Xuan Wei hospitals)	Case–control	Cases: 113 Controls: 107	Cases: 54.9 (M) Controls: 54.5 (M)	Cases: 65 Controls: 64			LC	Sputum		Age, type of current fuel use, sex, pack-years smoking, and lifetime smoky coal use	6

^1^ Cohort acronyms: NHS, Nurse’s Health Study; HPFS, Health Professionals Follow-Up Study; PLCO, Prostate Lung Colorectal and Ovarian Cancer Screening Trial; SWHS, Shanghai Women’s Health Study; ATBC, Alpha-Tocopherol Beta-Carotene Study; UPHS, University of Pennsylvania Health System. ^2^ Lung cancer acronyms: LC, Lung Cancer; SCLC, Small Cell Lung Cancer; NSCLC, Non-Small Cell Lung Cancer; LAUD, Lung Adenocarcinoma; LUSC, Lung Squamous Cell Carcinoma. ^3^ Newcastle–Ottawa Scale.

**Table 2 ijms-26-06610-t002:** Results of the stratified analysis of the LC risk estimates according to mtDNAcn.

	Combined Risk Estimate		Test of Heterogeneity			Publication Bias	
	N. ^b^	Value (95% CI)	Q	I^2^ %	*p*	*p* (Egger Test)	*p* (Begg Test)
ALL (5art) ^1^	33	0.94 (0.49–1.78)	132.48	96.98	0.00	0.93	1.00
Excluding Bonner et al. [38] ^2^	31	0.81 (0.39–1.67)	126.82	97.63	0.00	0.75	1.00
SEX							
M	4	1.13 (0.65–1.98)	10.63	71.77	0.014	0.51	0.17
F	4	1.36 (0.93–2.01)	4.75	36.84	0.19	0.35	0.50
Smoking Status							
CURRENT	4	1.37 (0.90–2.08)	5.38	44.25	0.15	0.18	0.17
EVER	5	1.10 (0.81–1.50)	5.21	23.30	0.27	0.62	0.33
NEVER	4	1.06 (0.76–1.48)	0.82	0.00	0.85	0.14	0.50

^1^ All articles are case–control studies. ^2^ In all articles, samples are peripheral blood. ^b^ The number of data used to calculate the risk.

## Data Availability

The data presented in this study are available upon request from the corresponding author.

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
