# Peer review of "Mitochondrial DNA Copy Numbers and Lung Cancer: A Systematic Review and Meta-Analysis"

_ijms, 2025, doi:10.3390/ijms26146610_

Round 1
Reviewer 1 Report
Comments and Suggestions for Authors
The manuscript reviewed and analyzed five studies exploring the relationship between mtDNA copy number and LC risk. The preliminary analysis showed that no significant correlation between mtDNA copy number and LC risk. As the manuscript showed, the conclusion is preliminary due to the limited numbers of studies. Nevertheless, this article still has some scientific significance.
Concerns:
1, The data from three areas were analyzed in the five studies. Were any differences among USA, China and Finland when considering the correlation between mtDNAcn and LC risk?
2, The data points and trend lines in Figure 4 should be properly annotated and explained in detail. The description of the figure is controversial between line 239 and line 259.
3, In line 204 and 218, the word “DNAm” should be “mtDNAcn”. In line 218 and 219, the title of 3.4 is repeated.
Author Response
REV 1
The manuscript reviewed and analyzed five studies exploring the relationship between mtDNA copy number and LC risk. The preliminary analysis showed that no significant correlation between mtDNA copy number and LC risk. As the manuscript showed, the conclusion is preliminary due to the limited numbers of studies. Nevertheless, this article still has some scientific significance.
Concerns:
1 The data from three areas were analyzed in the five studies. Were any differences among USA, China and Finland when considering the correlation between mtDNAcn and LC risk?
We thank the reviewer for this insightful question. Due to the limited number of included studies (n=5), subgroup analyses by country (USA, China, Finland) were not feasible without risking overinterpretation of sparse data. However, we observed that the effect estimates varied slightly across studies, with no clear or consistent pattern attributable to geographical location. For instance, studies conducted in the USA (Kim et al., Meng et al., Kennedy et al.) reported both null and slightly inverse associations, while the Chinese study (Bonner et al.) showed a higher LC risk associated with lower mtDNAcn, though with wide confidence intervals. The Finnish study (Hosgood et al.) also showed a non-significant association. Given the heterogeneity in study design, population characteristics (e.g., smoking prevalence, age, ethnicity), and mtDNAcn quantification methods, it is difficult to disentangle the effect of geographic location from other confounders.
In addition, current literature does not provide consistent evidence supporting significant geographic or ethnic differences in baseline mtDNA copy number. While some studies have reported modest variations associated with sex, age, or certain population-specific factors, there is no conclusive data indicating that mtDNAcn distribution systematically differs across countries or ethnic groups in a way that would explain the observed heterogeneity in LC risk. Nonetheless, we agree that this remains an important area for further investigation, especially considering the potential influence of environmental exposures, lifestyle, and genetic background. We have addressed this aspect in limitations sectino fo our study as follows:
“Moreover, the potential influence of geographic or ethnic variability in baseline mtDNA copy number cannot be excluded. Although current evidence does not support consistent differences across populations, further studies are needed to clarify whether contextual or population-specific factors may contribute to the observed heterogeneity.”
2 The data points and trend lines in Figure 4 should be properly annotated and explained in detail. The description of the figure is controversial between line 239 and line 259.
We thank the reviewer for highlighting this important point. We acknowledge that the description of Figure 4, particularly between lines 239 and 259, may have caused confusion due to inconsistent language and insufficient explanation of the plotted elements.
We have now revised the figure legend and the related text in the Results section to ensure clarity and consistency. Specifically, we have:
- Clearly defined the x-axis (standardized exposure levels corresponding to mtDNAcn quartiles) and y-axis (odds ratios for LC risk).
- Specified that the black solid line represents the overall dose-response trend derived from meta-regression.
- Indicated that the blue dashed lines represent the 95% confidence intervals around the estimated trend.
- Corrected the discrepancy in the text describing the direction of the trend (initially described as both decreasing and increasing) that it was due to a typo, and we truly apologize for this.
Added clarification that the slight inverse trend (non-significant) suggests a potential, though inconclusive, decrease in LC risk with increasing mtDNAcn.
These edits improve the interpretability of the figure and align the graphical representation with the narrative in the manuscript.
3 In line 204 and 218, the word “DNAm” should be “mtDNAcn”. In line 218 and 219, the title of 3.4 is repeated.
We thank the reviewer for pointing out. We have replaced “DNAm” with “mtDNAcn” in lines 204 as suggested. Furthermore, we have removed the duplicated title of section 3.4 found in lines 218.
Reviewer 2 Report
Comments and Suggestions for Authors
The systematic review and meta-analysis titled "Mitochondrial DNA Copy Number and Lung Cancer: A Systematic Review and Meta-Analysis" by Chiavarini et al. addresses a potentially valuable predictive biomarker for lung cancer risk.
The study is methodologically sound and contributes meaningfully to the current body of knowledge. With a few minor revisions, it will be suitable for dissemination to the scientific community.
Comments:
Abstract
- The abstract could be slightly more informative regarding the outcomes of the systematic review. In particular, it would benefit from including key quantitative details, such as the number of studies included and the total number of participants analyzed.
Introduction
No comments.
Materials and methods
- Please clarify whether data extraction and quality assessment were performed independently by at least two reviewers. Additionally, it would be helpful to specify the method used to resolve any discrepancies.
- I would suggest reconsidering the inclusion of the study that utilized a different source of biological material – namely, sputum – as opposed to blood.
Results
- Figure 1 does not contain number of articles searched from Scopus.
- In Figure 1 please specify the number of studies excluded for particular reasons.
- Since the original studies compared mitochondrial DNA copy number across quintiles, it would be valuable to present the specific cut-off values or ranges for these quintiles in Table 1.
Discussion
No comments.
General comments
- Please note that multiple references in the text should be cited, for example, as [1-3], not as [1], [2], [3].
- Also revise the structure of paragraphs headings as there are some minor mistakes and inconsistencies.
Author Response
REV 2
The systematic review and meta-analysis titled "Mitochondrial DNA Copy Number and Lung Cancer: A Systematic Review and Meta-Analysis" by Chiavarini et al. addresses a potentially valuable predictive biomarker for lung cancer risk.
The study is methodologically sound and contributes meaningfully to the current body of knowledge. With a few minor revisions, it will be suitable for dissemination to the scientific community.
Comments:
Abstract
- The abstract could be slightly more informative regarding the outcomes of the systematic review. In particular, it would benefit from including key quantitative details, such as the number of studies included and the total number of participants analyzed.
We thank the reviewer for this suggestion. We have revised the abstract to include the number of studies included in the meta-analysis as well as the total number of participants analyzed (3,748 participants across 5 studies). Additionally, we clarified that mitochondrial DNA copy number was measured in blood or sputum samples and compared across different quantiles.
The revised text now reads: “Five studies, including a total of 3.748 participants, met the eligibility criteria. MtDNA copy number was measured in blood or sputum samples and compared across different quantiles.”
Introduction
No comments.
Materials and methods
- Please clarify whether data extraction and quality assessment were performed independently by at least two reviewers. Additionally, it would be helpful to specify the method used to resolve any discrepancies.
Thank you very much for your valuable comment.
We confirm that the selection process was conducted independently by two blinded assessors. In cases of uncertainty or disagreement, a third assessor was consulted to reach consensus.
Additionally, data extraction was independently performed by two assessors. Their findings were subsequently compared and discussed to ensure completeness and accuracy. These procedures are consistent with the PRISMA guidelines and have been clarified accordingly in the revised manuscript.
In paragraph 2.3, Data Extraction and Quality Assessment, of the manuscript, we inserted “Two reviewers independently performed data extraction and quality assessment. Disagreements were resolved by consensus; a senior researcher was consulted when necessary to reach a final decision” before “For each selected study, we extracted the following information…”.
- I would suggest reconsidering the inclusion of the study that utilized a different source of biological material – namely, sputum – as opposed to blood.
We thank the reviewer for this valuable observation. We acknowledge that the study by Bonner et al. used sputum samples, which differ from the peripheral blood samples used in the other included studies. For this reason, we conducted a sensitivity analysis excluding this study to evaluate its potential impact on the overall results. As reported in the manuscript (Section 3.3), the exclusion of Bonner et al. did not substantially change the main findings, and the association between mtDNAcn and lung cancer risk remained non-significant (OR = 0.81; 95% CI: 0.39–1.67).
Given the limited number of eligible studies and the relevance of the Bonner et al. dataset in terms of sample size and methodological quality, we opted to include it in the main analysis while also clearly reporting the results of the analysis with and without it. This approach, in our view, strengthens the transparency and robustness of the meta-analysis.
Results
- Figure 1 does not contain number of articles searched from Scopus.
The number of articles retrieved from Scopus was 29. This information was already included in Figure 1, but it may not have been clearly visible due to a formatting issue that occurred when the document was downloaded.
- In Figure 1 please specify the number of studies excluded for particular reasons.
A total of 11 studies were excluded for specific reasons. This number was already reported in Figure 1, but it may not have been clearly visible for the same reason reported in point 4.
- Since the original studies compared mitochondrial DNA copy number across quintiles, it would be valuable to present the specific cut-off values or ranges for these quintiles in Table 1.
We thank the reviewer. For formatting reasons, we preferred not to add two additional columns with the cut-off values of the quintiles, as this would have further complicated and reduced the clarity of an already complex table.
Discussion
No comments.
General comments
- Please note that multiple references in the text should be cited, for example, as [1-3], not as [1], [2], [3].
We thank the reviewer. We have revised the manuscript to cite more references in the text using the format [1-3], as suggested.
- Also revise the structure of paragraphs headings as there are some minor mistakes and inconsistencies.
We thank the reviewer. We have revised the structure of paragraph headings to correct minor errors and ensure the consistency of the manuscript.
Round 2
Reviewer 1 Report
Comments and Suggestions for Authors
All my concerns are addressed and I have no more questions.